Molecular dynamics ensemble refinement of the heterogeneous native state of NCBD using chemical shifts and NOEs

Papaleo Elena 1 3
Camilloni Carlo 2 4
Teilum Kaare 1
Vendruscolo Michele 2
Lindorff-Larsen Kresten lindorff@bio.ku.dk 1
1 Structural Biology and NMR Laboratory, Linderstrøm-Lang Centre for Protein Science, Department of Biology, University of Copenhagen , Copenhagen , Denmark
2 Department of Chemistry, University of Cambridge , Cambridge , United Kingdom
3 Current affiliation:  Computational Biology Laboratory, Danish Cancer Society Research Center , Copenhagen , Denmark
4 Current affiliation:  Department of Biosciences, University of Milano , Milano , Italy
Rosano Camillo
Electronic publication date: 2018 Jul 4
Publication date: 2018
Volume: 6
Electronic Location ID: e5125
Received 2018 Mar 16; Accepted 2018 Jun 8
Copyright: ©2018 Papaleo et al.
Copyright year: 2018
Copyright holder: Papaleo et al.
License: This is an open access article distributed under the terms of the Creative Commons Attribution License, which permits unrestricted use, distribution, reproduction and adaptation in any medium and for any purpose provided that it is properly attributed. For attribution, the original author(s), title, publication source (PeerJ) and either DOI or URL of the article must be cited.
License URL: https://creativecommons.org/licenses/by/4.0/

Keywords: Protein structure, Protein dynamics, NMR, Molecular dynamics, Force field, Ensemble refinement, Maximum entropy

Funding: Novo Nordisk Foundation Danish e-Infrastructure Cooperation HPC Grant 2013 PRACE Research Infrastructure Resource Curie Elena Papaleo and Kresten Lindorff-Larsen were supported by a Hallas-Møller stipend from the Novo Nordisk Foundation (to Kresten Lindorff-Larsen). The project was also supported by the Danish e-Infrastructure Cooperation HPC Grant 2013 and the PRACE Research Infrastructure Resource Curie (France, 7th PRACE Tier0, NMRFUNC). The funders had no role in study design, data collection and analysis, decision to publish, or preparation of the manuscript.

==============================
Many proteins display complex dynamical properties that are often intimately linked to their biological functions. As the native state of a protein is best described as an ensemble of conformations, it is important to be able to generate models of native state ensembles with high accuracy. Due to limitations in sampling efficiency and force field accuracy it is, however, challenging to obtain accurate ensembles of protein conformations by the use of molecular simulations alone. Here we show that dynamic ensemble refinement, which combines an accurate atomistic force field with commonly available nuclear magnetic resonance (NMR) chemical shifts and NOEs, can provide a detailed and accurate description of the conformational ensemble of the native state of a highly dynamic protein. As both NOEs and chemical shifts are averaged on timescales up to milliseconds, the resulting ensembles reflect the structural heterogeneity that goes beyond that probed, e.g., by NMR relaxation order parameters. We selected the small protein domain NCBD as object of our study since this protein, which has been characterized experimentally in substantial detail, displays a rich and complex dynamical behaviour. In particular, the protein has been described as having a molten-globule like structure, but with a relatively rigid core. Our approach allowed us to describe the conformational dynamics of NCBD in solution, and to probe the structural heterogeneity resulting from both short- and long-timescale dynamics by the calculation of order parameters on different time scales. These results illustrate the usefulness of our approach since they show that NCBD is rather rigid on the nanosecond timescale, but interconverts within a broader ensemble on longer timescales, thus enabling the derivation of a coherent set of conclusions from various NMR experiments on this protein, which could otherwise appear in contradiction with each other.

Introduction

Molecular dynamics (MD) simulations have the potential ability to provide an accurate, atomic-level description of the conformational ensembles of proteins and their macromolecular complexes (Lindorff-Larsen et al., 2005; Dror et al., 2012; Perilla et al., 2015). Nevertheless, simulations are limited by both the accuracy of the physical models (force fields) and the precision due to conformational sampling (Mobley, 2012; Esteban-Martín, Bryn Fenwick & Salvatella, 2012). To overcome these problems, it is possible to bias the simulations using experimental data as structural restraints taking into account the inherent averaging in the experiments (Lindorff-Larsen et al., 2005; Camilloni et al., 2012; Lehtivarjo et al., 2012; Pitera & Chodera, 2012; Camilloni & Vendruscolo, 2014; Ravera et al., 2016). In this way, the experimental data can be included as a system-specific force-field correction, that combines the two sources of information using Bayesian statistics or the maximum entropy principle (Pitera & Chodera, 2012; Roux & Weare, 2013; Cavalli, Camilloni & Vendruscolo, 2013; Boomsma, Ferkinghoff-Borg & Lindorff-Larsen, 2014; White & Voth, 2014; Olsson et al., 2014; MacCallum, Perez & Dill, 2015; Hummer & Köfinger, 2015; Bonomi et al., 2016; Bonomi et al., 2017; Bottaro et al., 2018). Among the many techniques that can be used to probe structure and dynamics of proteins, NMR spectroscopy stands out as being able to provide a number of different parameters that are sensitive to protein dynamics over different timescales, as well as to probe the “average structure” in solution.

Previously, replica-averaged simulations have provided a wealth of information about the dynamical ensembles that proteins can attain in solution (Lindorff-Larsen et al., 2005; Tang, Schwieters & Clore, 2007; Fenwick et al., 2011; Camilloni et al., 2012; Ángyán & Gáspári, 2013; Camilloni, Cavalli & Vendruscolo, 2013a; Camilloni, Cavalli & Vendruscolo, 2013b; Islam et al., 2013; Vögeli et al., 2014; Camilloni & Vendruscolo, 2014). Exploiting improvements in the accuracy and speed of predicting protein NMR chemical shifts from protein structure (Kohlhoff et al., 2009; Han et al., 2011; Li & Brüschweiler, 2012), it is now possible to combine experimental chemical shifts with molecular simulations to study protein structure and dynamics (Wishart & Case, 2001; Cavalli et al., 2007; Shen et al., 2008; Wishart et al., 2008; Robustelli et al., 2009; Robustelli et al., 2010; Boomsma et al., 2014). In particular, chemical shifts can be used as replica-averaged structural restraints to determine the conformational fluctuations in proteins (Camilloni et al., 2012; Camilloni, Cavalli & Vendruscolo, 2013a; Camilloni, Cavalli & Vendruscolo, 2013b; Kannan et al., 2014; Kukic et al., 2014; Krieger et al., 2014). By using experimental data as a “system specific force field correction” (Boomsma, Ferkinghoff-Borg & Lindorff-Larsen, 2014) such experimentally-restrained simulations remove some of the uncertainty associated with imperfect force fields and sampling (Tiberti et al., 2015; Löhr, Jussupow & Camilloni, 2017).

Previously, we developed a dynamic-ensemble refinement (DER) approach for determining simultaneously the structure and dynamics of proteins by combining distance restraints from nuclear Overhauser effect (NOE) experiments, dynamical information from relaxation order parameters and MD simulations (Lindorff-Larsen et al., 2005). Similarly, it has been demonstrated that accurate ensembles of conformations that represent longer timescale dynamics can be obtained from residual dipolar couplings (Lange et al., 2008; De Simone et al., 2009; De Simone et al., 2015). These applications have, however, relied on a type of data (relaxation order parameters or residual dipolar couplings) that may not be readily available.

We therefore sought to extend this approach to study conformational variability using more commonly available data, thus making the DER method more generally applicable. We thus focus on using NMR chemical shifts and NOEs as these are both commonly available and are averaged over long, millisecond timescales. We demonstrate the potential by describing the structural heterogeneity of a highly dynamic protein. Our method relies on supplementing the sparse experimental data with the experimentally-validated CHARMM22* force field (Piana, Lindorff-Larsen & Shaw, 2011), which provides a relatively accurate description of the subtle balance among the stability of the different secondary structure classes, and which has been shown to provide a good description of many structural and dynamical aspects related to protein structure (Shaw et al., 2010; Lindorff-Larsen et al., 2012a; Lindorff-Larsen et al., 2012b; Piana, Lindorff-Larsen & Shaw, 2012; Papaleo et al., 2014; Rauscher et al., 2015). Our hypothesis was that using a more accurate force field would make it possible to determine an accurate ensemble from less information-rich experimental data. In particular, though chemical shifts in principle contain very detailed information, this information is difficult to extract using current methods.

As object of our study we selected NCBD (the Nuclear Coactivator Binding Domain) of CBP (CREB Binding Protein), a 59-residue protein domain that has been experimentally characterized in substantial detail. Experiments on NCBD have revealed a rich and complex dynamical behaviour of the protein in solution (Demarest et al., 2004; Ebert et al., 2008; Kjaergaard, Teilum & Poulsen, 2010; Kjaergaard, Poulsen & Teilum, 2012; Kjaergaard et al., 2013). For a protein of its size, NCBD displays surprisingly broad NMR peaks, suggestive of conformational heterogeneity with relatively slow interconversion between different states. Nevertheless, it was possible to assign both backbone and side chain chemical shifts and determine a number of conformationally-averaged inter-nuclear distances, including a few long-range contacts, via NOE experiments (Ebert et al., 2008; Kjaergaard, Teilum & Poulsen, 2010; Kjaergaard, Poulsen & Teilum, 2012). NMR relaxation experiments suggest that the protein, at least on the nanosecond timescale, is relatively rigid (Kjaergaard, Poulsen & Teilum, 2012). NCBD forms complexes with several other proteins, where it intriguingly folds into remarkably different tertiary structures (Demarest et al., 2002; Qin et al., 2005). For example, the structure of NCBD in complex with ACTR (Demarest et al., 2002) and certain other partners (Waters et al., 2006; Lee et al., 2010) resembles the average structure populated by NCBD in the absence of binding partners (Fig. 1), whereas the structure of NCBD is markedly different when bound to the protein IRF-3 (Qin et al., 2005). Thus, the dynamical properties of NCBD, and its ability to adopt different conformations, appear crucial for its diverse biological functions.

Figure 1 A previously determined structural model of the conformation of NCBD in solution.

The structure is shown as a cartoon (PDB entry: 2KKJ) with the protein coloured from the N- to the C-terminus (blue to red). The three α-helices are labelled. The goal of this work is to provide an ensemble of structures that represent the conformational fluctuations associated with this average conformation.

Our results show that a dynamic ensemble refinement that combines NOEs, chemical shifts and the CHARMM22* force field provides a rather accurate description of the structural dynamics of the ground state structure of NCBD. We show via cross-validation with independent NMR data that all three components (the two sources of experimental information and the force field) contribute to the overall accuracy. The ensemble that we obtained reveals a relatively broad distribution of conformations, reflecting the conformational heterogeneity of NCBD on the millisecond timescale. Further, we quantified the level of structural fluctuations that would be measured by relaxation experiments and demonstrate that, on the nanosecond timescale, NCBD is more rigid, thus helping to reconcile earlier conflicting views of this protein.

Materials and Methods

Ensemble generation

MD simulations were performed using Gromacs 4.5, (Pronk et al., 2013) coupled to a modified version of Plumed 1.3, (Bonomi et al., 2009) and using either the CHARMM22* (Piana, Lindorff-Larsen & Shaw, 2011) or CHARMM22 (MacKerell et al., 1998) force fields. As starting structure for most simulations we used the first conformer from a previously determined NMR structure of free NCBD as deposited in the PDB entry 2KKJ (Kjaergaard, Teilum & Poulsen, 2010). To evaluate the effect of our choice of the initial structure, we also performed one simulation starting from an alternative NCBD conformation (PDB entry: 1ZOQ, chain C) (Qin et al., 2005). Missing residues in 1ZOQ (compared to 2KKJ) were rebuilt by Modeller 9.11 (Fiser & Šali, 2003).

The protein was embedded in a dodecahedral box containing 8372 TIP3P water molecules (Jorgensen et al., 1983) and simulated using periodic boundary conditions with a 2 fs timestep and LINCS constraints (Hess et al., 1993). Production simulations were performed in the NVT ensemble with the Bussi thermostat (Bussi, Donadio & Parrinello, 2007) using a pre-equilibrated starting structure for which the volume was selected based on a short NPT simulation. NaCl was added to a concentration of ∼20 mM to reproduce the experimental conditions at which chemical shifts and NOEs were determined (Kjaergaard, Teilum & Poulsen, 2010). The van der Waals and short-range electrostatic interactions were truncated at 9 Å, whereas long-range electrostatic effects were treated with the particle mesh Ewald method (Essmann et al., 1995).

We carried out MD simulations with replica-averaged experimental restraints using 1, 2, 4 or 8 replicas (Table S1 gives an overview of the simulations that were performed). The use of replica-averaged restrained simulations enables us to use different equilibrium experimental observable as a restraint in MD simulation in a way that minimises the risk of over restraining because replica-averaging is a practical implementation of the maximum entropy principle. As a control we also performed a simulation that was not biased by any experimental restraints (i.e., an unbiased simulation). To examine the role played by each of the different types of experimental data, we also performed simulations in which we included different combinations of the experimental restraints: chemical shifts only (CS), NOEs only (NOE), and both chemical shifts and NOEs (CS-NOE). In the simulations, each replica was evolved through a series of simulated annealing (SA) cycles between 304 and 454 K for a total duration of 0.6 ns per cycle. Specifically, for each SA cycle we performed: (i) 100 ps at 304 K, (ii) a linear increase of the temperature from 304 to 454 K over 100 ps, (iii) 100 ps at 454 K, and (iv) a linear cooling from 454 K to 304 K in the remaining 300 ps. Each new cycle was initiated from the final structure from the previous cycle. We only used structures from the 304 K portions of the simulations for our analyses, corresponding also to the temperature at which the NMR data were recorded (Kjaergaard, Teilum & Poulsen, 2010). Example scripts for performing the simulations are available as supporting information.

Chemical shifts for the backbone atoms (Cα, C′, Hα, H and N) and Cβ CS (deposited in BMRB entry 16363) were used as restraints (with the exception of the Cβ of glutamines, which we have sometimes found to be imprecisely predicted). The resulting dataset includes 54 Cα, 37 Cβ, 52 Hα and 48 C′, H and N chemical shifts, respectively. The backbone chemical shifts cover most of the NCBD sequence with the exception of the first four to six N-terminal residues, depending on type of chemical shifts. The Cβ chemical shifts for the first seven N-terminal and last five C-terminal residues, as well as for some residues of the loops connecting the α-helices, are missing with few exceptions.

During the structure determination protocol, chemical shifts were calculated by CamShift (Kohlhoff et al., 2009) for all the nuclei for which an experimental value is available and then averaged over the replicas. The resulting average over the replicas was compared with the experimental value, and the ensemble as a whole restrained using a harmonic function with a force constant of 5.2 kJ mol−1ppm−2 (Camilloni et al., 2012; Camilloni, Cavalli & Vendruscolo, 2013a). At the higher temperatures, T, explored during the simulated annealing, the force constant was scaled by a factor of (304 K/T). The value of the force constant was chosen roughly to match the calculated chemical shifts to experiments within the uncertainty of the CamShift predictor; the experimental uncertainty of the chemical shifts is negligible in comparison.

NOE restraints were obtained by 455 NOE-derived distance intervals (Kjaergaard, Teilum & Poulsen, 2010) (BMRB entry 16363) of which 46 were long-range (i.e., separated by more than 4 residues). The proton–proton distances, r, were calculated and averaged as r−6 over the replicas (Tropp, 1980; Lindorff-Larsen et al., 2005). We used a flat-bottomed harmonic function implemented in Gromacs to restrain the calculated averaged distances within the experimentally-derived intervals. We used a variable force constant for the NOE-restraints during the SA cycles, allowing the protein to sample more diverse structures in the high-temperature regime and thus to decrease the risk of getting trapped in local minima. Force constants of 1,000, 20 and 125 kJ mol−1 nm−2 were used for the 304 K phase, a heating phase (from 304 K to 454 K) and cooling phase (from 454 K to 304 K), respectively.

In short, in the replica-averaged simulations we calculated at each step and for each replica-conformation the atomic distances that were measured by the NOE experiments and the backbone chemical shifts. These calculated single-conformer values were then averaged (linearly for the shifts and using r−6 averaging for the distances) to determine the replica-averaged values, which were then compared to the experimentally determined values. Thus, the simulations penalize deviations between the calculated ensemble averages and experimental values but allow fluctuations of individual structures. In this way, the simulations are biased so as to agree with the experimental data as a whole, while allowing individual conformations to take on conformations whose NMR parameters differ from the experimentally derived averages.

To examine the role of the force field used in our approach, we compared the results from two different force fields belonging to the same family (CHARMM). These force fields mostly differ for the main-chain dihedral angle potential, as well a few parameters for certain side chains. Specifically, we used either the CHARMM22* (Piana, Lindorff-Larsen & Shaw, 2011) or CHARMM22 (MacKerell et al., 1998) force fields. The CHARMM22* force field is a refined version of CHARMM22 that includes modified backbone torsion angles optimized to give improved agreement with a range of NMR data in simulations of peptides of various lengths and secondary structure propensities. Furthermore in a previous comprehensive evaluation of protein force fields it, was demonstrated that these two force fields resulted in very different levels of agreement between simulations and experiments (Lindorff-Larsen et al., 2012a), making it possible for us to evaluate the importance of force field accuracy in restrained simulations.

Unbiased simulations for the calculation of fast-timescale order parameters

We also performed 28 independent unbiased MD simulations, each 50 ns long, at 304 K and with the same computational setup as the restrained simulations, but without any restraints. As starting points, we selected seven different structures from each of the four replicas obtained in the CS-NOE-4 ensemble (Table S1). In particular, the seven structures were selected from the SA cycles after convergence (i.e., at SA cycles 65, 75, 85, 95, 100, 110, 125). We calculated fast timescale order parameters, which correspond to those measured by NMR relaxation measurements, from these 28 unbiased simulations using a previously described approach (Maragakis et al., 2008). In particular, we calculated bond-vector autocorrelation functions (independently from each simulation) including both internal motions and overall tumbling of NCBD. The resulting correlation functions were then averaged over the 28 simulations and subsequently fitted globally to a Lipari-Szabo model (Lipari & Szabo, 1982) to yield relaxation order parameters. To calculate order parameters that report on the long-timescale motions we first aligned the full ensemble and then calculated order parameters as ensemble averages (Maragakis et al., 2008).

Analyses of convergence and cross validation

We used two different methods to examine the convergence of our simulations. First, we used the ENCORE ensemble comparison method (Lindorff-Larsen & Ferkinghoff-Borg, 2009; Tiberti et al., 2015) to quantify the overlap between the structural ensembles. The latter is based on clustering the structures using affinity propagation (setting the “preference value” in the clustering to 12) and subsequent comparison of the ensembles by calculating the Jensen–Shannon (JS) divergence between pairs of ensembles by comparing how they populate the different clusters. For additional details, please confer to original descriptions of the method (Lindorff-Larsen & Ferkinghoff-Borg, 2009; Tiberti et al., 2015). As an alternative method, we calculated the Root Mean Square Inner Product (RMSIP) over the first 10 eigenvectors obtained from a principal component analysis of the covariance matrix of atomic (Cα-atoms) fluctuations (Amadei, Linssen & Berendsen, 1993).

To cross-validate our ensembles we calculated the chemical shifts of side chain methyl hydrogen and carbon atoms using CH3Shift (Sahakyan et al., 2011) (both 1H and 13C shifts) and PPM (Li & Brüschweiler, 2012) (only 1H shifts) and compared to the previously determined experimental side chain chemical shifts. In particular, we compared the calculated side chain chemical shifts with the experimental values (deposited in BMRB entry 16363) using a reduced χ2 metric. In this metric, the square deviation between the calculated and experimental values were normalized by the variance of the chemical shift predictor (for each type of chemical shift) and the total number of chemical shifts, so that low numbers indicate good agreement between experimental and calculated chemical shifts.

Figure 2 Assessment of the convergence of the simulations.

The similarity between structural ensembles was quantified using structural clustering with Affinity Propagation and subsequent comparison of the ensembles by Jensen–Shannon (JS) divergence. The JS divergence between two identical ensembles is zero, and it has previously been found that values less than 0.3 represent similar ensembles. We monitored the evolution of the JS-divergence in two different tests, either by comparing two replicas from the same simulation (i.e., CS-NOE-2, black) or two simulations with the same force field and restraints but different starting structures (i.e., CS-NOE-2 starting from 2KKJ and 1ZOQ structures, respectively, grey). As described in the text we discarded the first 45 SA cycles before calculating the ensemble similarity for the test with different starting structures.

Results

Convergence of the simulations

Before assessing the accuracy of the different structural ensembles that we generated, we first ensured that the simulated annealing protocol allowed us to obtain converged ensembles that represent the dynamical properties encoded in the experimental restraints and the molecular force field. To quantify convergence of the ensembles, we calculated two different measures of the overlap between the subspaces sampled by different simulations.

First, we used a previously described approach (Lindorff-Larsen & Ferkinghoff-Borg, 2009; Tiberti et al., 2015), which is based on a quantification of the extent to which the different ensembles mix during conformational clustering, to calculate the Jensen–Shannon (JS) divergence between the ensembles (Fig. 2). A JS divergence of zero is evidence of identical ensembles, and it has previously been observed that a JS divergence in the range of 0.1–0.3 represents similar ensembles (Lindorff-Larsen & Ferkinghoff-Borg, 2009; Tiberti et al., 2015). We expect that in a converged replica-averaged simulation that the different replicas should populate equally the different structural basins. With this in mind, we calculated the JS divergence between two replicas in a simulation restrained by NOEs and chemical shifts (Fig. 2, black line). We find that after approximately ∼30 cycles of simulated annealing the two replicas have covered approximately the same conformational space with the JS divergence stabilizing around 0.2–0.3 with the fluctuations in the JS-divergence representing the stochastic nature of the simulations. Thus, we decided to discard the first 45 simulated annealing cycles from all the simulations. As an alternative measure of ensemble similarity we also calculated the Root Mean Square Inner Product (Hess, 2002) (RMSIP) with very similar results. In particular, the similarity of the two replicas converge to an RMSIP value greater than 0.83 (here RMSIP = 1 is expected for fully overlapping ensembles).

As a second, perhaps even more stringent, test of convergence we also examined whether two simulations with the same number of replicas and experimental restraints, but initiated from substantially different starting structures, converge to similar ensembles. Indeed, we find that simulations initiated from two distinct structures of NCBD (Table S1) converge to similar ensembles when the first 45 cycles are discarded as initial equilibration (Fig. 2, grey line). Thus, based on these two tests we concluded that our sampling protocol allows us to obtain structural ensembles that represent the force field and restraints employed.

Assessment of the accuracy of the NCBD ensembles

Once we had assessed the convergence of the simulations, we analysed the different ensembles to evaluate their accuracy. To do so, we back-calculated experimental parameters that were not used as restraints and compared them with the experimental values. As our different simulations employed different sets of experimental restraints, not all experimental data can be employed for validation purposes. For example, while the NOEs can be used to evaluate the quality of an ensemble obtained using CS-restraints, they can obviously not be used to validate an ensemble that was generated using those NOEs as restraints.

We first examined whether the CS or NOE restraints alone are sufficient to increase the accuracy in the description of the conformational ensemble of NCBD. We thus compared unbiased simulations with simulations biased by either CS or NOEs by cross-validation with the measured NOEs and CS, respectively.

We back-calculated NOEs from the inter-proton distances and observed substantial violations (some greater than 2 Å) in both unbiased and CS ensembles (Fig. S1) independently of the number of replicas used for the averaging. To determine the origin of these discrepancies we calculated intramolecular contacts between side chains, and observed an overall decrease in these (from 27 in the previously-determined NMR ensemble, to 14 and 17 in unbiased and CS-restrained, respectively). More specifically we found a loss of inter-helical contacts between helices α1 and α2 in the simulations, in agreement with our finding of several long-range NOEs that are violated in these ensembles.

These results demonstrate that the CS-restraints and MD force field, as implemented here, are not sufficient to provide a fully accurate description of the conformational ensemble of NCBD. Similarly, we found that back-calculation of backbone chemical shifts from the unbiased simulation and, to a lesser extent a NOE-restrained ensemble, resulted in deviations from experiments. We therefore decided to determine conformational ensembles that combine the information of the NOEs, chemical shifts and force field in replica-averaged simulations (CS-NOE) aiming to provide a more accurate structural ensemble of NCBD than possible via the application of just one of the two classes of restraints. We also assessed the influence of the choice of force field since we expected that a more accurate ensemble could be obtained with the relatively limited amounts of experimental data when using a more accurate force field. Thus, we compared simulations using either the CHARMM22 force field (CS-NOE-4-C22 simulation), or a more recent and accurate force field variant, CHARMM22* (CS-NOE simulations).

As both the NOEs and backbone chemical shifts were used as restraints they cannot be used for validation of these ensembles. Instead, we turned to side-chain methyl chemical shifts for a comparison and validation of the different ensembles. Methyl-containing residues, for which the chemical shifts are available, cover the entire protein structure and are thus excellent probes of both local structure (13C methyl chemical shifts, which are mostly dependent on the rotameric state) and long-range contacts (1H methyl chemical shifts). The methyl chemical shifts were predicted by CH3Shift (Sahakyan et al., 2011) and the resulting values compared to experiments, separating the contributions from 13C and 1H. We then calculated χred2 thus taking into account the inherent uncertainty of the chemical shift predictions (Sahakyan et al., 2011).

As also indicated by the calculation of NOEs and backbone chemical shifts, we find that the side chain chemical shifts predicted from the unbiased simulation (green line in Fig. 3) deviates substantially from experiments. The introduction of backbone chemical shift restraints (CS ensembles, orange line in Fig. 3) provides a better structural ensemble than the force field alone, especially for 13C methyl chemical shifts and when averaged over 2 or 4 replicas. We also calculated the chemical shifts from NOE-derived ensembles, obtained with or without replica-averaging. Surprisingly, we find that the ensembles obtained using NOEs as replica-averaged restraints (NOE, magenta line in Fig. 3) perform slightly worse than the CS ensemble. Thus, when evaluated in this way, ensembles derived by MD refinement using either backbone chemical shifts or NOEs do not increase accuracy compared to the ensemble deposited in the PDB.

Figure 3 Validation of the structural ensemble using side-chain methyl chemical shifts.

We calculated the deviation between experimental and predicted (A) 13C and (B) 1H side-chain chemical shifts from each MD ensemble. The results are shown as a function of the number of replicas used for the averaging of the simulations. The previously determined NMR structure (black) and unbiased MD simulation (green) do not involve replica averaging and are shown as horizontal lines.

By combining the NOEs, chemical shifts and the CHARMM22* force field we were, however, able to obtain even more accurate ensembles, in particular when averaging over four replicas, as assessed by the ability to predict side chain 13C and 1H methyl chemical shifts (Fig. 3). Interestingly we find that not only the experimental data but also the CHARMM22* force field contributes to the improved agreement with the experimental data. Indeed, when we employ both chemical shift and NOE-based restraints in simulations averaged over 4 replicas, but replacing the CHARMM22* force field by an earlier, less accurate variant of the same force field (CHARMM22; CS-NOE-4-C22) (Lindorff-Larsen et al., 2012a) we find that the accuracy decreases dramatically. Calculations of 1H methyl chemical shifts using PPM (Li & Brüschweiler, 2012) instead of CH3Shift demonstrate that the conclusions are robust to the method for calculating the chemical shifts (Fig. S2). Similarly, calculations of the chemical shifts using the ensemble generated from the alternative starting structure (CS-NOE-2-1ZOQ) resulted in essentially the same agreement with the experimental data as when simulations were initiated from the 2KKJ structure (Fig. 3), confirming the conclusions from the convergence analysis described above (Fig. 2). The CS-NOE-4 ensemble, which we found to provide the most accurate representation of the free state of NCBD in solution, is shown in Fig. 4. It is a relatively broad ensemble of conformations, where the three helical regions are maintained overall, but differ in the lengths and relative positions of the three α-helices.

Figure 4 Conformational ensemble of the free state of NCBD obtained by molecular dynamics simulations with the CHARMM22* force field and replica-averaged CS and NOE restraints.

The α-helices are represented as cylinders and the structural ensemble was aligned using THESEUS.

Small Angle X-ray scattering (SAXS) measurements have been carried out for NCBD in solution (Kjaergaard, Teilum & Poulsen, 2010) and previously been compared to simulation-derived ensembles of NCBD (Knott & Best, 2012; Naganathan & Orozco, 2013). We thus calculated the radius of gyration (Rg) using CRYSOL (Svergun, Barberato & Koch, 1995) for the various ensembles. In all cases we find that the average Rg values are in the range of 13.7 Å–14.9 Å. These values are comparable to that obtained previously from simulations (13.7 Å) (Knott & Best, 2012) but lower than the values estimated from a Guinier analysis of the experimental data (∼16.5 Å) or an ensemble-optimization method (18.8 Å) (Kjaergaard, Teilum & Poulsen, 2010). We note, however, that the experimental values also include contributions from a ∼8% population of unfolded protein that is not captured by our simulations. Although a detailed understanding is lacking for the role of solvation on the SAXS properties of partially disordered proteins we, however, expect that the discrepancy between experiment and simulation should be ascribed to remaining force field deficiencies. Indeed, overly large compaction of proteins is a common problem of most atomistic force fields (Piana, Klepeis & Shaw, 2014) though recent work suggests that, at least for fully disordered proteins, that modified protein-water interactions can improve accuracy (Nerenberg et al., 2012; Best, Zheng & Mittal, 2014; Henriques, Cragnell & Skepö, 2015; Mercadante et al., 2015; Piana et al., 2015). We also note that while the force field used here (CHARMM22*) in certain cases has been shown to produce too compact structures, (Piana et al., 2015) in other cases it appears to perform quite well (Rauscher et al., 2015). We expect that resolving these issues will require both further force field developments (Best, 2017) as well as improved methods for comparing experiments and SAXS experiments (Hub, 2018).

A unified view of NCBD dynamics

While the broad peaks and sparse NOEs are suggestive of a rather dynamic protein, previous NMR relaxation measurements of side chain dynamics found relatively high order parameters (Srelaxation2) comparable to values found in well-ordered proteins (Kjaergaard, Poulsen & Teilum, 2012). To shed light on this apparent discrepancy and to assess whether our relatively broad structural ensemble is compatible with mobility on different timescales, we calculated S2 values representing different timescales.

To mimic the dynamics probed in relaxation experiments we selected 28 structures from each of the four replicas of the CS-NOE-4 ensemble sampled at seven different SA steps. Starting from each of these conformations we performed 50 ns of unbiased MD simulation (in total 1.4 µs, Fig. S3), and from each simulation we calculated the autocorrelation functions of the N-H bond vectors (without removing the overall rotational motion of the protein). These correlation functions were subsequently averaged and fitted to the Lipari-Szabo model to estimate the Srelaxation2 values, which report on the nanosecond dynamics of the protein (Fig. 5, black line). The results show a relatively rigid ensemble on the ns timescale attested by high order parameters throughout most of the polypeptide backbone.

Figure 5 Calculation of order parameters from MD simulations to probe short and long timescale dynamics.

We calculated S2 order parameters that reflect either motions faster than overall tumbling of the protein (black) or longer timescale motions that give rise to chemical shift and NOE averaging (red). For reference, the main chain Root Mean Square Deviation (RMSD) values of the 28 unbiased simulations that we used to calculate the Srelaxation2 values are shown in Fig. S3.

To quantify the backbone dynamics on the longer timescales that may influence both the NOE and chemical shifts (but which the relaxation measurements would not be sensitive to) we defined and calculated “Schemicalshift2”-values from the structural variability in the ensemble after aligning the structures. These S2 values include contributions also from any millisecond-timescale motions that might be present in the ground state of NCBD. As internal and overall motions cannot be decoupled, the results of such calculations will depend on how the ensemble is aligned. In our calculations we chose theseus (Theobald & Steindel, 2012) as the least biased method to align the structures (Fig. 4). These order parameter calculations reveal a broader distribution of conformations with additional, longer-timescale dynamics evident both in loop regions and the C-terminal region, even though relatively high S2 values are found in the regions of secondary structures (Fig. 5, grey line).

A similar analysis of side chain motions suggests even greater differences in motions present on relaxation and chemical shift timescales. In particular, we find that, for methyl-bearing side chains, Schemicalshift2-values are on average lower than Srelaxation2-values by 0.4 compared to an average difference of 0.2 for the backbone amides. Finally, we note that although both calculated Schemicalshift2-values and Srelaxation2-values correlate strongly with the experimentally determined side chain Srelaxation2-values (Spearman correlation coefficient of 0.9 and 0.8, respectively), a more quantitative analysis is hampered by several issues including: (i) the presence of a small population of unfolded protein in the experiments, (ii) the difficulty in appropriate model selection of the calculated correlation functions, (iii) the well-known observation of too-fast rotational motions of proteins in the TIP3P model that we used and (iv) uncertainties in the parameterization of the rotational motions in the experimental analyses. We note, however, the potential complications that arise from the fact that the Schemicalshift2-values were obtained from simulations with an experimental bias, whereas the Srelaxation2-values were obtained from simulations starting from such a biased ensemble, but performed with the standard CHARMM22* force field.

Discussion

We have performed restrained simulations of the small protein NCBD and find that after approximately ∼30 cycles of simulated annealing that two “identical” replicas have covered approximately the same conformational as judged by the JS divergence between them. Similarly, we find that simulations initiated from two distinct structures of NCBD converge to similar ensembles when the first 45 cycles are discarded. Thus, based on these two tests we concluded that our sampling protocol allows us to obtain structural ensembles that represent the force field and restraints employed.

Once we had assessed the convergence of the simulations, we analysed the different ensembles to evaluate their accuracy. As our different simulations employed different sets of experimental restraints, not all experimental data can be employed for validation purposes.

Our results revealed that the CS-restraints and MD force field, as implemented here, are not alone enough to describe accurately the conformational ensemble of NCBD. We therefore determined conformational ensembles that combine the information of the NOEs, chemical shifts and force field, and validated them using side-chain methyl chemical shifts. The results show that by combining the NOEs, chemical shifts and the CHARMM22* force field we are able to obtain even more accurate ensembles (compared to using these data individually), in particular when averaging over four replicas. Thus, we find that the CS-NOE-4 ensemble provides the most accurate representation of the free state of NCBD in solution among the different ensembles we have studied. We, however, find that this ensemble is slightly more compact than expected from experiment, and suggest that a more careful analysis of the SAXS data and a force field that gives a better balance between compact and expanded structures are necessary to solve these issues.

Our results also shed new light on the amount and time-scales of the dynamics in NCBD. In particular, our calculations of order parameters demonstrate that NCBD may be described as a semi-rigid protein on fast-timescales, but with additional dynamics in the backbone and–in particular–side chains on timescales longer than the rotational correlation time of the protein, as also previously suggested (Kjaergaard, Poulsen & Teilum, 2012).

Conclusions

We have presented an application of the dynamic-ensemble refinement method to study the native state dynamics of NCBD. In the original implementation of DER we combined NMR relaxation order parameters with NOEs in MD simulations (Lindorff-Larsen et al., 2005). This approach was here extended to the combination of chemical shifts and NOEs to make it more generally applicable. In particular, our results show that it is possible to combine NOEs, backbone chemical shifts and an accurate MD force field into replica-averaged restrained simulations, and that all three components add substantially to the accuracy of the resulting NCBD ensemble.

NMR structures are typically obtained by combining distance information from NOE measurements with in vacuo simulations, in certain cases with subsequent refinement by short, MD simulations in explicit solvent. Further, the inherent ensemble averaging of the experimental data is typically not exploited explicitly. In this way, standard NMR structures can provide highly accurate models of the “average structure” of a protein, but only little information about the conformational heterogeneity around this average.

Replica-averaged MD simulations make it possible to obtain structural ensembles that match the experimental data according to the principle of maximum entropy (Pitera & Chodera, 2012; Roux & Weare, 2013; Cavalli, Camilloni & Vendruscolo, 2013; Boomsma, Ferkinghoff-Borg & Lindorff-Larsen, 2014; White & Voth, 2014; Olsson et al., 2014). In such calculations prior information, here in the form of a molecular mechanics force field, is biased in a minimal fashion to agree with the experimental data. Thus, to obtain an accurate ensemble, such simulations require an accurate force field, an efficient sampling approach as well as sufficient experimental information. Our results show that, at least in the case of the small, but relatively mobile protein NCBD, it is possible to perform such simulations when NOEs are supplemented by the information available in the backbone chemical shifts and a well-parameterized molecular force field. The application of the experimentally-derived structural restraints helps overcome at least some of the deficiencies in force field accuracy and also improves sampling of the relevant regions of conformational space. While we find that four replicas are optimal for the system and data studied here, we expect that this value might vary between systems and hence recommend evaluating it, e.g., by comparing to independently measured data such as the side chain shifts analysed here.

Our approach also allowed us to probe the structural heterogeneity arising from both short- and long-timescale dynamics by the calculation of order parameters. In the case of NCBD we found that this protein can be described as a relatively rigid protein domain on a fast timescale, as attested by the high relaxation order parameters that, nevertheless, displays additional motions in both the backbone and side chains on longer timescales. This situation is reminiscent of the molten globule state of apomyoglobin, that also displays restricted motions on the nanosecond timescale but with greater motions on a slower timescale (Eliezer et al., 2000; Meinhold & Wright, 2011). The current study also provides the groundwork for further studies on NCBDs intricate conformational dynamics, and the relationship to ligand binding (Dogan et al., 2012; Zijlstra et al., 2017). Given the importance of understanding and quantifying protein dynamics, in particular on long timescales, we expect that our approach, which uses only commonly available data, and possible combined with novel algorithms for enhancing sampling (Bonomi et al., 2016; Bonomi, Camilloni & Vendruscolo, 2016), will have a wide range of applications.

Supplemental Information

Table S1 Summary of the MD ensembles obtained in this study

The number of simulated annealing (SA) cycles listed represents the number of cycles performed (for each replica) and analyzed after discarding the first 45 cycles for convergence.

Click here for additional data file.

Figure S1 Cross-validation of unbiased and CS-restrained ensembles using NOE measurements

We calculated the total number of NOE violations in the unbiased and CS-restrained ensembles. We used 455 NOE-derived distance restraints (BMRB entry 16363) of which 409 are short- and 46 are long-range (i.e., separated by more than 4 residues). In addition to the total number of violations (all), we also separated the violations into different categories depending on their magnitude (0.5Å–1Å, 1Å–2Åor greater than 2Å) and whether they are short (0–4 residues apart) or long-range (more than 4 residues apart).

Click here for additional data file.

Figure S2 Validation of the structural ensemble using 1H methyl chemical side-chain chemical shifts

We calculated the deviation between experimental and calculated side-chain 1H chemical shifts from each MD ensemble with PPM to compare with the CH3Shift predictions reported in Fig. 3B. The results are shown as a function of the number of replicas used for the averaging of the simulations. The previously determined NMR structure (black) and unbiased MD simulation (green) do not involve replica averaging and are shown as horizontal lines.

Click here for additional data file.

Figure S3 Main chain RMSD of the 28 NCBD unbiased simulations started from conformations extracted from the CS-NOE-4 ensemble

Click here for additional data file.

Data S1 Raw data and input files

This file contains input files for running the molecular simulations, and the resulting ensemble of NCBD obtained using NOEs and chemical shifts.

Click here for additional data file.

We would like to thank Magnus Kjaergaard, Wouter Boomsma, Matteo Tiberti and Peter Wright for fruitful discussion and comments.

Additional Information and Declarations

Competing Interests

Author Contributions

Data Availability

Elena Papaleo is as an Academic Editor for PeerJ.

Elena Papaleo performed the experiments, analyzed the data, contributed reagents/materials/analysis tools, prepared figures and/or tables, authored or reviewed drafts of the paper, approved the final draft.

Carlo Camilloni, Kaare Teilum and Michele Vendruscolo analyzed the data, contributed reagents/materials/analysis tools, authored or reviewed drafts of the paper, approved the final draft.

Kresten Lindorff-Larsen conceived and designed the experiments, analyzed the data, contributed reagents/materials/analysis tools, authored or reviewed drafts of the paper, approved the final draft.

The following information was supplied regarding data availability:

The raw data are provided in a Supplemental File.

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
