# Peer review of "Molecular dynamics ensemble refinement of the heterogeneous native state of NCBD using chemical shifts and NOEs"

_PeerJ, doi:10.7717/peerj.5125_

## Round 0.1 · original submission · Minor Revisions

The paper is very well written and scientifically sounds. However, both the referees found some minor points to be amended. Please find their comments attached

Reviewer 1 ·

Basic reporting

I would better detail the simulated annealing protocol (see comments below).

Experimental design

no comment

Validity of the findings

no comment

Additional comments

The work by Papaleo and coworkers uses experimental restraints (typically
averaged on long times and multiple conformations) to force,
in an average sense, molecular dynamics simulations to be consistent with them.
Average restraints allow the system to explore
conformational space thus providing realistic simulations.
The approach is assessed by comparing the results with experimental data
not used as restraints in simulation.
The results show that the approach is valid and they also further support the validity
of the refined force field charmm22*.

The work is well written and arranged in a very logical way.

I find only the methods section not very detailed. I imagine readers
will like to implement the same protocol in their own simulations.
To this end the protocol should be detailed and, possibly, tools made
available.
In particular the SA protocol should be detailed:
"In the simulations, each replica was evolved through a series of simulated
annealing (SA) cycles between 304 and 454K for a total duration of 0.6 ns per
cycle. We only used structures from the 304K portions of the simulations for
our analyses."
it is not clear exactly what is the protocol, is the temperature dropped
linearly or non-linearly at each step, or in discrete steps?
The cycles at higher temperature start from the lowest temperature structure
from the previous cycle?
Could the authors detail better the protocol in order to make the simulation
fully repeatable by the readers?

- Why are the Cbetas of glutamines not considered?

- I do not understand the phrase "averaged over the replicas of the replicas"

Although this is reported in the cited publication, due to the important role
in the simulations I would also report the major differences of charmm22* with
respect to charmm22 with the cmap correction.

·

Basic reporting

The article is well-written. The cited literature is accurate.

In Figure 3 , the blue dots are described in the legend as "CS-NOE-C22*", but according to Table S1 it should be "CS-NOE-4-C22". Indeed the CHARM22 ff is used for these simulations.

Experimental design

This research fits within Aims and Scope of PeerJ. The article provides sufficient information for others to adopt the same methodological approach for different protein systems.

NMR chemical shifts have a measurable T-dependence, especially for amides (see Baxter, N.J. & Williamson, M.P. J Biomol NMR (1997) 9: 359. https://doi.org/10.1023/A:1018334207887). I presume that CamShift does not take this into account, so that all chemical shift predictions during the MD runs are effectively at 304 K (or 298 K, which is not relevant within the uncertainty of the prediction). The authors may want to mention this explicitly.

Validity of the findings

No comment

Additional comments

Simulations with 4 replicas appear to reproduce the data better than those with 8 replicas. Can the authors comment on this? It could be useful to researchers who would like to apply this methodology to other proteins.

---

## Round 0.2 · accepted · Accept

The authors amended the paper accordingly with the referees' comments.

# Reviewer 1 ·

Basic reporting

All my comments have been completely addressed.

Experimental design

All my comments have been completely addressed.

Validity of the findings

All my comments have been completely addressed.

Additional comments

All my comments have been completely addressed.

·

Basic reporting

The authors have addressed the points raised by the reviewers in a satisfactory manner.

Experimental design

no comment

Validity of the findings

no comment